# The Intricate Role of Ecdysis Triggering Hormone Signaling in Insect Development and Reproductive Regulation

**DOI:** 10.3390/insects14080711

**Published:** 2023-08-16

**Authors:** Pooja Malhotra, Saumik Basu

**Affiliations:** Department of Entomology, Washington State University, Pullman, WA 99164, USA; pooja.malhotra@wsu.edu

**Keywords:** Inka cell, ecdysis-triggering hormone, 20-hydroxy ecdysone, juvenile hormone, fecundity, corazonin, JHAMT

## Abstract

**Simple Summary:**

Ecdysis triggering hormone (ETH) synthesized and secreted from peripheral Inka cells on the tracheal surface play a vital role in the triggering and orchestration of ecdysis in insects and other arthropod species. Binding of ETH to the ETH receptor (ETHR) located on the peptidergic neurons in the central nervous system (CNS) facilitates synthesis of ecdysis-related neuropeptides. Although the role of ETH on ecdysis has been well-studied in various holometabolous insects, only very little information is available in hemimetabolous insects. Recent studies in hemimetabolous insects have identified and characterized diverse features of ETH precursors, ETH and ETH receptors. In addition to this, the involvement of ETH in Juvenile hormone (JH) mediated courtship short-term memory (STM) retention and long-term courtship memory regulation and retention has also been shown recently. Here, we extensively studied ETH signaling cascades in various insects and their roles in triggering diverse functions in the adults and juvenile insects including development and reproductive regulation. Understanding the intricate details of ETH signaling in diverse insect species could lead to the identification of novel targets and chemicals, as well as the development of various sustainable pest management solutions.

**Abstract:**

Insect growth is interrupted by molts, during which the insect develops a new exoskeleton. The exoskeleton confers protection and undergoes shedding between each developmental stage through an evolutionarily conserved and ordered sequence of behaviors, collectively referred to as ecdysis. Ecdysis is triggered by Ecdysis triggering hormone (ETH) synthesized and secreted from peripheral Inka cells on the tracheal surface and plays a vital role in the orchestration of ecdysis in insects and possibly in other arthropod species. ETH synthesized by Inka cells then binds to ETH receptor (ETHR) present on the peptidergic neurons in the central nervous system (CNS) to facilitate synthesis of various other neuropeptides involved in ecdysis. The mechanism of ETH function on ecdysis has been well investigated in holometabolous insects such as moths *Manduca sexta* and *Bombyx mori*, fruit fly *Drosophila melanogaster*, the yellow fever mosquito *Aedes aegypti* and beetle *Tribolium castaneum etc. In contrast,* very little information is available about the role of ETH in sequential and gradual growth and developmental changes associated with ecdysis in hemimetabolous insects. Recent studies have identified ETH precursors and characterized functional and biochemical features of ETH and ETHR in a hemimetabolous insect, desert locust, *Schistocerca gregaria*. Recently, the role of ETH in Juvenile hormone (JH) mediated courtship short-term memory (STM) retention and long-term courtship memory regulation and retention have also been investigated in adult male *Drosophila*. Our review provides a novel synthesis of ETH signaling cascades and responses in various insects triggering diverse functions in adults and juvenile insects including their development and reproductive regulation and might allow researchers to develop sustainable pest management strategies by identifying novel compounds and targets.

## 1. Introduction

The exoskeleton or cuticle provides various benefits to the insects including muscular support, desiccation tolerance, maintenance of shapes and appearances, enhanced protection against predators and more strength against preys. Hence, molting (i.e., shedding of the exoskeleton) in insects occur every time they want to grow. Shedding of the older cuticle between developmental stages is an evolutionarily conserved mechanism among many invertebrate species including nematodes and arthropods, especially in insects. The sequence of events for ecdysis has been described well in insects [1,2]. During ecdysis in insects, the new cuticle is formed by shedding the old cuticle following a series of programmed events marked as the onset of a new instar. Ecdysis behavior in insects takes place in three consecutive phases known as, pre-ecdysis, ecdysis and post-ecdysis under control of the skeletal muscle contraction and regulated by peptide hormones and neurotransmitters [3,4,5]. The shedding of old cuticles occurs during pre-ecdysis (highly variable among insects) through abdominal movement, followed by loosening of the old cuticular layer [1]. During ecdysis, the removal of the loosened cuticular layer takes place through peristaltic contraction, increased cardiac activity and increased body size resulting from increased absorption of air [2]. During post-ecdysis the newly formed cuticle undergoes expansion, hardening and tanning [6,7].

Our understanding of the mechanism of endocrine control of ecdysis (pre-ecdysis, ecdysis and post-ecdysis) comes from studies performed on mostly holometabolous insects, such as, moths *Manduca sexta* and *Bombyx mori* [8,9,10,11], fruit fly *Drosophila melanogaster* [12,13,14], and the beetle, *Tribolium castaneum* [15]. *Four hormones were identified to control various ecdysis sequences:* Pre-ecdysis triggering hormone (PETH), Ecdysis triggering hormone (ETH), Eclosion hormone (EH), Bursicon and Crustacean cardioactive peptide (CCAP) [16]. Other important factors were also implicated to have significant roles in controlling ecdysis in insects, including Kinins, myoinhibitory peptides FMRFamide [10,14], and Corazonin (CRZ) [10,14,17,18,19].

Sequential events of ecdysis involve both PETH and ETH, neuropeptides released from endocrine Inka cells which function directly on the central nervous system (CNS) to initiate ecdysis in moths and fruit flies [20]. Ecdysis triggering hormone (ETH) was first identified in the hemolymph of the tobacco hornworm *M. sexta* where it was found to undergo substantial changes during metamorphosis and function as a trigger of ecdysis [21]. Inka cells were found to be the major source for ETH production which are present in the epitracheal glands on the tracheal surface and was first described in the silk moth, *Bombyx mori* [22]. Later, PETH, another related peptide, and similar peptides (ETH1 and ETH2) were also identified in epitracheal gland extracts [8,23]. These paired epitracheal glands have been found near spiracles in the thoracic and abdominal surfaces in most of the lepidopteran and dipteran species as well as some coleopteran and hymenopteran species. 

During ecdysis, ETH is secreted from tiny Inka cells distributed throughout the tracheal system in some other holometabolous and all other hemimetabolous insects reported so far [23,24,25]. Recent studies showed that the release of ETH from endocrine Inka cells is regulated by both 20-hydroxy ecdysone (20E, an ecdysteroid hormone that directly regulates the expression of both ETH and ETH receptors from Inka cells), EH (another photo-periodically regulated brain hormone), and CRZ (neuropeptides from the brain and corpora cardiaca) [18,26]. Increased 20E levels induce expressions of genes encoding *eth* in Inka cells and *ethr* in the CNS. Reduced 20E levels function as releasing signals of CRZ from the brain-corpora cardiaca (CC)-corpora allata (CA) complex and enhances secretion of PETH/ETH from Inka cells. PETH/ETH action on various targets in the CNS results in the initiation and orchestration of various phases of ecdysis in insects [27].

Genes encoding ETH and ecdysis triggering hormone receptors (ETHR, ETHR-A, and ETHR-B) were identified and well-characterized in various holometabolous insects, such as moths *M. sexta* and *B. Mori,* fruit fly *D. melanogaster,* red flour beetle *Tribolium castaneum,* mosquito *Aedes aegypti,* oriental fruit fly *Bactrocera dorsalis*, and in a few hemimetabolous insects, such as pea aphid *Acyrthosiphon pisum*, and desert locust *Schistocerca gregoria* [4,8,9,15,20,21,24,25,28,29,30,31]. In the desert locust, *Schistocerca gregaria*, the ETHR receptor (SchgrETHR) exhibits dual coupling properties of both cyclic adenosine mono phosphate (cAMP) and Ca^2+^ (two second messengers) by increasing their levels, when activated by both SchgrETH1 and SchgrETH2 [25]. Ecdysis triggering hormone receptors are seven transmembrane (TM) rhodopsin-like G-protein coupled receptors in different insects, ticks, and crustaceans [24]. Selective RNAi knockdown of ETHR-A and ETH, but not ETHR-B, in *B. dorsalis* caused developmental abnormalities by interfering with different phases of larval ecdysis [31].

In this current review, we investigated the molecular and biochemical details of ETH and ETHR, mechanistic details of various complex neuroendocrine signaling networks involved in diverse functions in both juvenile and adult insects, and variations of signaling responses among various insects.

## 2. Structure, Function, and Regulation of ETH and ETHRs

Ecdysis triggering hormone (ETH) is a linear amidated peptide comprised of 11–34 amino acids. C-terminal amidation (a post-translational modification) of ETH plays significant roles in the biological functions of this hormone [2,28]. Ecdysis triggering hormones identified from insects and crustaceans share a conserved C-terminus, but their N-terminus varies from species to species [32]. In *M. sexta,* the *eth* gene consists of three exons and two introns. ETH mRNA encodes 114 amino acid polypeptides (including 22 amino acid signal peptides), which undergo processing to give rise to 11 amino acid PETH, 26 amino acid ETH, and another 47 amino acid ETH-associated peptides [8]. Therefore, cDNA and the gene-encoding ETH also encode a novel peptide PETH (pre-ecdysis triggering hormone), which plays a crucial role in pre-ecdysis [8]. The upstream regulatory region of the *eth* gene contains direct repeats of the ecdysone receptor response element, which implicate the possible regulatory roles of ecdysone over ETH secretion. The presence of several other complex response elements in the promoter region of the *eth* gene confirms the possibility of additional factors in regulating ETH synthesis and release [8]. Small Inka cells were identified as sites for *eth* expression and are scattered throughout the tracheal system in hemimetabolous and some holometabolous insects. In contrast, 8–9 pairs of large Inka cells were present in both thoracic and abdominal epitracheal segments [23]. In *M. sexta,* a single Inka cell produces 2 pmol PETH/ETH, while this amount is increased to 8–10 pmol in pharate pupae [8,23]. In *M. sexta*, the concentration of PETH/ETH was found to be at 20–35 nmol in hemolymph during larval ecdysis, while in pupa, this level is slightly high at 30–45 nmol [8,33].

The synthesis of ETH from the *eth* gene is highly regulated by peak levels of ecdysteroid through interaction with its 20E receptor on Inka cells (EcR) and a basic leucine zipper (bZIP) gene, cryptocephal (crc) [34]. The secretion of ETH is under strong control of EH and coazonin (CRZ) [17,35]. The receptor for ETH (ETHR) is a G-protein-coupled receptor (GPCR), identified in various insects, including fruit fly, *D. melanogaster* [29,36] and in *M. sexta* [14]. Two subtypes of ETHR, ETHR-A and ETHR-B, were produced through alternative splicing and differentially expressed in peptidergic and aminergic neurons of the CNS, frontal ganglion, and corpora allata (CA, an endocrine gland that produces juvenile hormone) under the regulation of increased ecdysteroid titer in both *Drosophila* and *Manduca*. Various studies identified orthologs of ETHR-A and ETHR-B in other holometabolous insects, including, e.g., *Tribolium castaneum, Bombyx mori* and *Aedes aegypti* [11,15,37]. Roller et al. also analyzed genomic sequences of both ETHR subtypes of various insect types (both holo- and hemimetabolous) ‘in silico’ and found conserved structures and organization of both subtypes of ETHRs among all the experimental holometabolous and hemimetabolous insects used in this study [24]. The same study also identified ETHR receptor orthologs in other arthropod species, including the Arachnida (e.g., tick *Ixodes*) and Crustacea (e.g., water flea *Daphnia*), indicating the ubiquitous role of ETH-ETHR signaling across diverse organisms undergoing ecdysis [24].

## 3. Role of ETH in the Regulation of Larval Ecdysis in Holo- and Hemimetabolous Insects

Increasing levels of the ecdysteroid hormone enhance the expression of *eth* genes from Inka cells and stimulate secretions of PETH and ETH and the episodic expression of ETHR both in peptidergic neurons and CA. Immediately after ecdysis, ETH loses its sensitivity toward CNS until the next ecdysone peak due to the loss of ETHR transcripts and internalization of ETHR proteins together with ligands. This ecdysis behavior in insects can be broadly categorized into three phases of variable duration, based on the insect size, age, sex and other environmental conditions [38]. The first phase is known as pre-ecdysis (consisting of pre-ecdysis I & II with durations from minutes to an hour) during which the older cuticle loses its connection with underlying muscles through the initiation of dorso-ventral contractions and air filling up the tracheal chamber. The next phase, ecdysis, is highly variable (from minutes to hours) and involves the shedding of older cuticle through peristaltic longitudinal contractions. The last phase, post-ecdysis, can also last from minutes to hours, and sees the new cuticle is stretched, hardened, tanned, and wings inflate while proteins in the new cuticles are cross-linked [1,23]. Various studies demonstrated eclosion hormone (EH) to be the principal hormone required for triggering ecdysis behaviors in larval insects [23,39]. In the light of recent research, it is evident that the regulation of ecdysis is a highly specific genetically programmed event and under the control of coordinated interaction of several hormones (mainly three peptide hormones including ETH, EH and CCAP), which further increase the degree of complexity in this process [40,41]. Similar to ETH, the synthesis of many hormonal peptides requires elevated levels of ecdysone, while their release require declining concentrations [40]. 

Release of ETH from Inka cells is under the regulation of neurohormones (EH and CRZ) and fluctuating concentrations of the ecdysteroid 20E [8,17,42]. 20-hydroxy ecdysone directly regulates the expression of both ETH and ETH receptors [8,26]. At the onset of sequential events of ecdysis behavior (e.g., 8 h before ecdysis in *M. sexta*), ETH levels and epitracheal Inka cell sizes enhance when 20E reaches peak concentrations. The release of ETH from Inka cells is triggered by both declining concentrations of 20E and secretion of brain neuropeptide CRZ [34,40]. ETH released from Inka cells can also activate peptidergic neurons in the CNS to produce Kinins and Corticotropin Releasing Factor (CRF)-like peptide to trigger pre-ecdysis [25]. Corazonin, a cardio-acceleratory peptide in insects, also triggers ecdysis and has been reported to be produced in the lateral neurosecretory cells of the brain and is released by neurons of the ventral ganglion [43]. Corazonin, even at very low concentrations, interacts with CRZ receptors (CRZRs) present on the surface of Inka cells to release ETH [17,19]. 

At the end of pre-ecdysis, ETH interacts with the receptors located on the neurosecretory cells (NSCs) in the brain to initiate the synthesis and release of EH, a photo-periodically regulated hormone, and is necessary for the circadian molting behavior in many insects [1,44]. CZR can release low levels of ETH from Inka cells to the hemolymph and forms a positive feedback loop with EH, which triggers a massive release of both hormones in the hemolymph mediated through elevated levels of Guanosine 3',5'-cyclic monophosphate (cyclic GMP/cGMP) [2,45]. The declining ecdysone (20E) titer following molting is not only important for ETH release but also for its sensitivity to CNS. High levels of EH together with elevated ETH concentration (positive feedback) is necessary for the maintenance of the pre-ecdysis phase in many insects including *D. melanogaster* and *M. sexta* [5,42]. This EH facilitates the release of crustacean cardioactive peptide (CCAP, a cardio-acceleratory peptide first identified in crustaceans (e.g., crabs) by activating the cells in the ventral nerve cord and together they terminate pre-ecdysis and further initiate ecdysis. CCAP triggers motor program necessary for ecdysis and shuts off program for pre-ecdysis (Figure 1, Table 1). CCAP is released from neurons on the ventral nerve cord in response to ETH and EH release [2]. When added to the CNS preparation lacking a brain from insects, it triggers the motor program necessary for ecdysis and terminates pre-ecdysis triggered by ETH [16]. 

CCAP also acts on nerves from the frontal ganglion that innervate muscles in the foregut and dilate them for the swallowing of air to generate pressure for shedding of the older larval exoskeleton [46]. Timely clearance of the airway during ecdysis occurs through the ETH-mediated peptide-signaling cascade, which involves neuropeptide kinin in *Drosophila* larvae [17]. The completion of ecdysis is also regulated by CCAP through the activation of motor patterns and plasticization of the cuticle, which further triggers the release of bursicon from neurons of sub-esophageal, thoracic, and the first segment of the abdominal ganglion, and triggers post-ecdysis through tanning of the larval cuticle (Figure 1, Table 1). During post-ecdysis, bursicon is present as a heterodimer (Bur/pBur) encoded by two genes, *bur* (bursicon) and *pbur* (product of bursicon) in order to carryout hardening and cuticle tanning, respectively. This behavior is necessary for new cuticle expansion and the Bur/pBur complex was found to bind to receptors on target cells [15]. Most importantly, the periodic ecdysis behavior during each molt involves the signaling network of various hormones, including ETH and through the coordinated expression of various genes associated with this process.

## 4. Coordinated Endocrine Network of ETH Promotes JH Production and Reproductive Success in Adult Insects

Besides its role in controlling the orchestrated ecdysis behavioral sequences in the juvenile phase of insects, the coordinated hormonal network of ETH is responsible for the production of juvenile hormone and promoting reproductive success in adult insects [11,26,47]. JH is a sesquiterpenoid hormone, synthesized and released by CA of insects and regulates various morphogenetic and reproductive functions [48,49]. The synthesis and release of JH from CA is under the control of several neuronal and hormonal factors [50]. Reduction in JH levels either through rapid degradation or programmed cell death in the CA gland is essential for adult development in insects [51,52].

The functional significance of JH in reproductive functions was thoroughly investigated by several researchers [53,54,55]. The involvement of ETH in the endocrine network for promoting reproductive functions was indicated by the presence of Inka cells in the tracheal system of adult insects [12,26]. The distribution of Inka cells differs greatly between *Drosophila* larvae and adults [12]. Unlike the even distribution of larval small Inka cells throughout the tracheal system, a more strategic distribution of 8–9 pairs of Inka cells were reported in adults. Two pairs are located in the thorax, of which one is situated at the anterior portion close to the CA, while four out of the seven pairs remain in the posterior part of the abdomen, near the reproductive organs [56]. The possibility of ETH involvement in JH production and synthesis was further supported through its interaction with ETHR expressed on the surface of CA in various holometabolous insects, such as the silkworm, *B. mori* [11], yellow fever mosquito, *A. aegypti* [47], and fruit fly, *D. melanogaster* [26]. In the yellow fever mosquito, *A. aegypti*, ETH regulates the expression of ETH receptors (*Aea*ETHRs) on CA and ensures time-dependent JH biosynthesis in pharate adults through Ca^+2^-dependent activation of the key enzyme JHAMT, without altering its genetic expression [47]. Activation of CA at the pupal stage through ETH trigger resulted in elevated JH production. Knocking down the expressions of ETHRs in the pupal stage through RNAi interferes with JH biosynthesis in newly emerged reproductive loss-of-function phenotypes of adult female *A. aegypti* [47].

The reproductive losses are characterized by smaller ovaries, lower fecundity, reduced expression and deposition of egg proteins in adult females, and reduced male reproductive potentials [47]. Normal ovary size, fecundity, and male reproductive potential can be rescued by injecting JH analog methoprene, an agonist of the JH receptor in various insects [47,57]. Besides RNAi knockdown of either ETH or ETHR during the adult stage, disruption of ETH signaling in the octopaminergic neurons innervating female reproductive tracts or ablation of Inka cell functions, can also be correlated with the ablation of total CA by reducing JH levels, and are responsible for reduced egg production, ovary size, and yolk deposition [26,58,59,60,61]. In *Drosophila*, a balance between 20E and JH controls oogenesis, and ETH plays a crucial role in maintaining this balance for successful oogenesis [26,62]. Together with *B. mori* and *A. aegypti*, the obligatory allatotropin function of ETH (i.e., regulating JH biosynthesis and reproductive success) was also reported in *D. melanogaster* [11,26,47]. Similar to larval stages, a strong regulatory influence of 20E on both ETH and ETHR was reported in adult insects [63,64]. ETH synthesis from Inka cells triggered by 20E are essential in regulating JH levels necessary for the maintenance of the normal size of the ovary, egg production, deposition of yolk proteins in mature oocytes, as well as fecundity in adult female and normal reproductive potential in adult male *D. melanogaster* [26]. Therefore, the 20E, ETH, and JH hormonal triad network are crucial for reproductive success in several holometabolous insects, including adult *D. melanogaster* males and females. Various environmental stress responses, silencing of ETHR on CA through RNAi or conditional Inka cell ablation hamper reproductive success in adult *D. melanogaster* [26,65].

The role of ecdysteroids, ETH and JH have also been well characterized in various developmental pathways of holometabolous insects [24,40,66] where bouts of steroids 20E surge and steroid ebb are responsible for periodic molting and ecdysis. The stimulation of ETH synthesis from Inka cells in response to 20E surge occurs with the help of transcription factor crc and EcR-B2 located on the Inka cell surface [67]. Peak levels of 20E suppress transcription of βFtz-F1, an orphan nuclear receptor necessary for secretory competence of ETH from Inka cell [34]. βFtz-F1 undergoes de-repression following a sharp decline in 20E and induces the release of ETH from Inka cells primed with secretary competency. Although the same mechanism is followed in adult insects, the fluctuations of 20E levels and bouts of 20E to trigger the synthesis and release of ETH for reproductive functions are yet to be investigated. Very little is known about the role of ETH in regulating JH biosynthesis in hemimetabolous adult insects. Only recently, low levels of *Schgr*ETHR and the precursor of *Schgr*ETH were reported from the corpora allata-corpora cardiaca (CA-CC) complex of the desert locust, *Schistocerca gregaria* [25,68].

The coordinated hormonal network of 20E, ETH, and JH in promoting reproductive success in adult holometabolous insects was well illustrated as shown in Figure 2. Ecdysone, synthesized from dietary cholesterol in insect prothoracic gland, is released in hemolymph and oxidized to 20E in peripheral tissues, such as fat bodies [69]. A surge of 20E activates synthesis and release of ETH via activation of EcR located on the surface of Inka cells of insect tracheal system and triggers the expression of ETHR in adult tissues, such as CA. Knockdown of EcR on Inka cell surfaces in adult flies through RNAi reduces female fecundity, egg production, vitellogenesis, and male reproductive potential and can be rescued by methoprene, a JH biosynthetic pathway analog. ETH synthesized and released from Inka cells then interacts with ETHRs located on the surface of cells in CA to promote elevated JH production, leading to increased egg production and fecundity in females and improved reproductive potential in male insects. Circulating levels of JH in the hemolymph exerts inhibitory effects on 20E production in adult insects, reported in several earlier reports [70,71,72]. Taken together, coordinated functional networks among 20E, ETH, and JH through sequential steps contribute to reproductive success in adult insects.

ETH also contributes to the JH biosynthesis and activation of the key enzyme JHAMT of the JH biosynthetic pathway through mobilization of Ca^+2^ from the intra- and extracellular storage and ensures proper developmental timing in pharate adult insects, which is dependent upon ETHR expression [47]. The binding of ETH to ETHR (a 7TM, GPCR) occurs on the cell surface in the CA activate G-protein associated with ETHR. The ETHR can activate the associated G protein by exchanging the bound GDP with a GTP. The α subunit of the G-protein, together with the bound GTP, then dissociates from the βγ subunits and further interferes with the downstream intracellular signaling. Here, the α subunit activates membrane-bound phospholipase C (PLC) that increases synthesis of inositol triphosphate (IP3) and diacyl glycerol (DAG) from PIP3. IP3 binds to IP3-R located on the endoplasmic reticulum (ER), intracellular storage of Ca^+2^, and releases stored Ca^+2^ in the cytoplasm. Ca^+2^ is also mobilized into the cytoplasm through store-operated Ca^+2^ entry channels (SOC) in response to ETH (ligand) binding to ETH receptors. An increase in the cellular levels of Ca^+2^ results in the activation of phosphokinase C, which in turn enhances JHAMT activity, ultimately leading to increased production and release of JH. Therefore, the activation of ETH signaling through the binding of ETH with ETHR on the surface of the CA cell is critical for the mobilization Ca^+2^ from storage. The increased level of Ca+2 in turn enhances JH biosynthesis through the activation of JHAMT, a key enzyme of the JH biosynthetic pathway (Figure 3). Besides reproductive success in adult insects, ETH also plays an essential role in the retention of short-term memory (STM) during an early adult critical period. The ETH-JH-DA cascade plays a crucial role in male *Drosophila* courtship short-term memory retention during early adulthood during a critical period in early adulthood (Figure 2). Similarly, ETH also plays a vital role in the regulation of long-term memory by regulating JH levels in *Drosophila* males either by influencing dopaminergic levels by JH or activating (MB) γ lobe neuropils in the insect brain through the convergence of both ETH and JH hormonal cascades [60,73]. Therefore, the coordinated endocrine network of the 20E-ETH-JH hormonal triad contributes to reproductive success in adult insects and courtship memory (both STM and LTM) retention and regulation in adult *Drosophila* males.

## 5. Conclusions and Future Direction

Current research suggests that ETH plays two major roles in insects. During juvenile phase, ETH regulates ecdysis behavioral sequences, while in adults, it is responsible for timely production of JH for promoting reproductive success [11,74,75]. ETH regulates JH biosynthesis by interacting with ETHRs located on the cells of CA by altering intracellular Ca^+2^ concentration or through activation of JHAMT (Figure 3). Several studies provided evidence for ETH to function as a master regulator of ecdysis in insects [16,24,47]. Although the synthesis and release of ETH under fluctuating concentrations of 20E were reported in both juvenile and adult developmental phases, the exact patterns are yet to be investigated in adult insects. ETH is crucial for retention of short-term courtship memory by regulating JH levels in adult *Drosophila* males. JH acts through dopaminergic neurons and influences them for the retention of courtship STM during a critical period. The ETH-JH-DA cascade is also important for retention of courtship STM in the early adult *Drosophila* male. The silencing of ETH receptors reduces JH levels and interferes with courtship memory [73]. Besides this, ETH plays a vital role in the modulation of memory in *Drosophila,* both directly in the mushroom body (MB) γ lobe neuropils and octopaminergic dorsal-anterior-lateral (DAL) neurons in the insect brain through Ca^2+^-mediated mechanisms and indirectly by regulating the JH production [60]. Taken together, the convergence of both ETH and JH hormonal cascades in the MB γ lobe, a functionally distinct area in the brain, plays a crucial role in the courtship long-term memory regulation [60]. Investigating similar functions of ETH in other insects and identifying the involvement of insect ETH signaling in controlling other essential pathways would be a future target of ETH signaling research.

Improvement of high throughput next generation sequencing, gene knockdown (RNAi), genome editing approaches (CRISPR/Cas9 genome editing system), and structural determination of various components of ETH signaling would be crucial for identifying various unknown components of complex ETH signaling networks in different insects and developing future pest control strategies. For example, ETHR (GPCR), a surface receptor of ETH signaling, can be used as an excellent target candidate for developing next-generation sustainable pest management strategies by selecting many compounds that target cell surface receptors [25]. More in-depth studies of various components and associated complex networks of ETH signaling are also required in many other species, including the hemimetabolous insects, to understand the intricate mechanistic details and differences in regulating ecdysis behavior and in the correct temporal sequence with respect to holometabolous insects (Appendix A).

## Figures and Tables

**Figure 1 insects-14-00711-f001:**
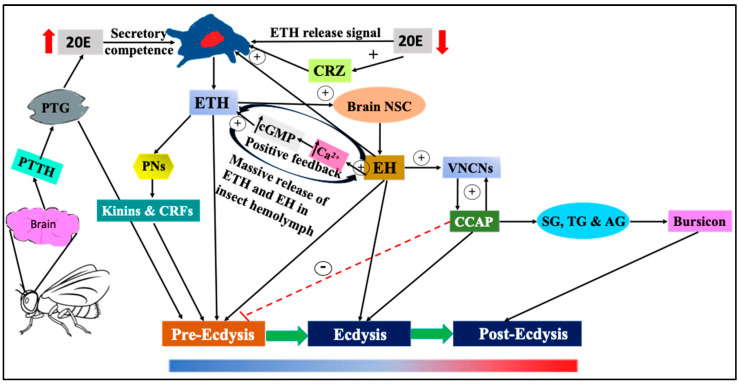
Complex network of ETH signaling in regulating ecdysis behavioral sequences in insect larvae. The release of ETH is under the control of various complex neurohormonal networks including 20E, EH and CRZ etc. 20E released from PTG in response to brain derived PTTG and when the concentration reaches its peak, it elicits the levels of both ETH in Inka cells and EH in brain. Releases of both ETH and EH are triggered by declining concentrations of ETH. Another highly conserved neuropeptide hormone, CRZ, produced in the lateral neurosecretory cells of the brain and are released by neurons of the ventral ganglion, even at very low concentrations, interacts with CRZ receptors (CRZRs) present on the surface of Inka cells to release ETH. ETH then activates peptidergic neurons in the CNS to produce kinins and corticotropin releasing factor (CRF)-like peptide to trigger pre-ecdysis. High levels of EH together with elevated ETH concentration (positive feedback) is necessary for the maintenance of pre-ecdysis and is mediated through high levels of cGMP. This EH facilitate the release of CCAP by activating the cells in the ventral nerve cord and together they terminate pre-ecdysis and further initiate ecdysis. CRZ triggers ecdysis along with EH and CCAP. The completion of ecdysis is also regulated by CCAP, which further triggers the release of Bursicon from neurons of SG, TG and AG and triggers post-ecdysis. Therefore, periodic ecdysis behavior during each molt involves the signaling network of various neuro-hormones and through coordinated expression of various genes associated with this process. PTTH = prothoracicotropic hormone; PTG = prothoracic gland; PNs = peptidergic neurons; 20E = 20-hydroxy ecdysone; ETH = ecdysis triggering hormone; NSC = neurosecretory cells; EH = eclosion hormone; VNCNs = ventral nerve cord neurons; CCAP = crustacean cardioactive peptide; cGMP = cyclic GMP; SG, TG and AG = sub-esophageal ganglia, thoracic ganglia and abdominal ganglia.

**Figure 2 insects-14-00711-f002:**
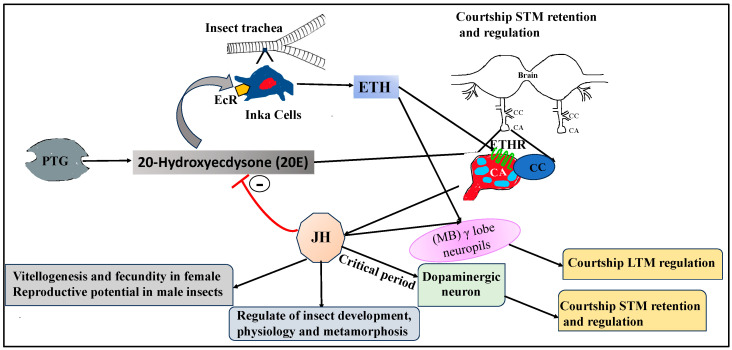
The coordinated endocrine network of the 20E-ETH-JH hormonal triad contributes to reproductive success in adult insects and courtship memory (both STM and LTM) retention and regulation in adult *Drosophila* males. A surge of 20E activates the synthesis and release of ETH via activation of the 20E receptor (EcR) located on the surface of Inka cells and triggers the expression of ETHR on CA. ETH synthesized and released from Inka cells then interacts with ETHRs located on the surface of cells in CA to promote elevated JH production, leading to increased egg production and fecundity in females and improved reproductive potential in male insects besides regulating insect development, physiology, and metamorphosis. Circulating levels of JH in the hemolymph exert inhibitory effects on the production of 20E in adult insects. Taken together, coordinated functional networks among 20E, ETH, and JH through sequential steps contribute to the reproductive success in adult insects and courtship memory (both STM and LTM) retention and regulation in adult *Drosophila* males. PG = prothoracic gland; CA = corpora allata; CC = corpora cardiaca; ETH = ecdysis triggering hormone; ETHR = ecdysis triggering hormone receptor; JH = juvenile hormone; EcR = 20E receptor; DA = Dopaminergic neuron; STM = short-term memory; LTM = long-term memory; MB = mushroom body.

**Figure 3 insects-14-00711-f003:**
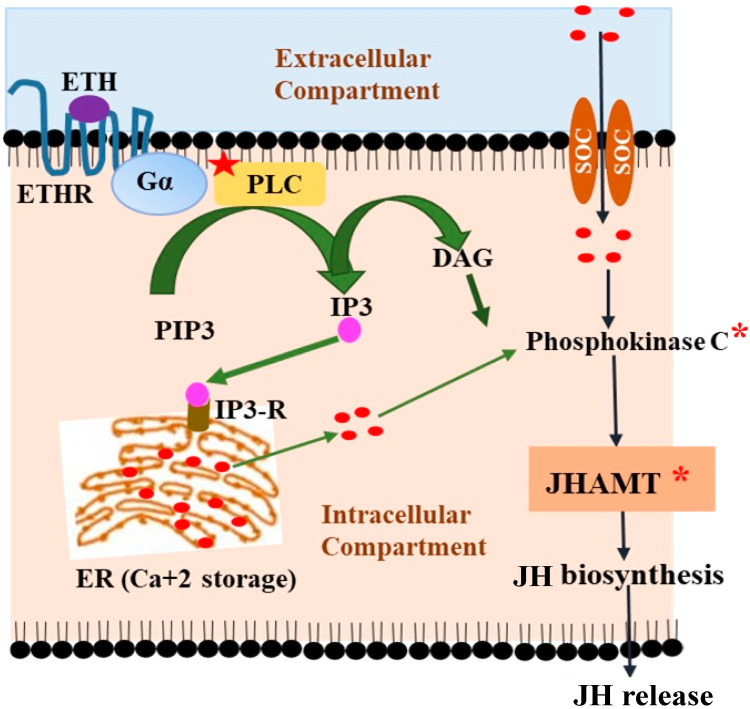
The role of Ca^+2^ mobilization through ETH-mediated signaling is responsible for the JH biosynthesis through the activation of JHMAT in CA. ETH plays an important role in the JH biosynthesis and activation of JHAMT, a key enzyme in the JH biosynthetic pathway through mobilization of Ca^+2^ from the intra- and extracellular storages. ETH binds to ETHR (a 7TM, GPCR), located on the cell surface in the CA, and activates the G-protein associated with the ETHR by exchanging the bound GDP with a GTP. The α subunit of the G-protein, together with the bound GTP, then dissociates from the βγ subunits to activate membrane-bound phospholipase C (PLC). PLC then catalyzes the conversion of inositol triphosphate (IP3) and diacyl glycerol (DAG) from PIP3. IP3 binds to the IP3-R located on the endoplasmic reticulum (ER), an intracellular storage of Ca^+2^, and releases stored Ca^+2^ in the cytoplasm. Ca^+2^ is also mobilized into the cytoplasm through the store-operated Ca^+2^ entry channels (SOC) in response to ETH (ligand) binding to ETH receptors. An increase in cellular levels of Ca^+2^ resulted in the activation of phosphokinase C, which in turn enhanced the JHAMT activity, ultimately leading to increased JH production and release. ETH = ecdysis triggering hormone; GPCR = G-protein coupled receptor; PIP3 = phosphatidylinositol 3,4,5-trisphosphate; IP3 = inositol triphosphate; IP3R = inositol triphosphate receptor; PLC = phospholipase C; PKC = phosphokinase C; DAG = diacylglycerol; SOC = store-operated Ca^2+^ entry channel; ER = endoplasmic reticulum; and JH = juvenile hormone; * = Activation [44], modified.

**Table 1 insects-14-00711-t001:** Hormones (both peptide- and neurohormones) released from glands and nerves/cells involved in various phases of ecdysis.

Phases of Ecdysis	Hormones	Gland and Nerves/Cells Involved
Pre-ecdysis (I and II)	PTTH, ETH (II), EH, 20E, PETH(I), Kinins and CRFs, myoinhibitory peptides FMRFamide	PTG, epitracheal Inka cells (Trachea), neuro secretary cells of brain, and peptidergic neurons in the CNS.
Ecdysis	ETH, EH and CCAP, CRZ	PTG, epitracheal Inka cells (Trachea), neuro secretary cells of brain, ventral nerve cord neurons, lateral brain neurosecretory cells projecting to the corpora cardiaca-corpora allata (CC-CA) complex, and in neurons of the ventral nerve cord.
Post-ecdysis	Bursicon	Neurons of sub-esophageal, thoracic, and first segment of abdominal ganglion.

## Data Availability

Data are contained within the article.

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
