# Peer review of "The Intricate Role of Ecdysis Triggering Hormone Signaling in Insect Development and Reproductive Regulation"

_insects, 2023, doi:10.3390/insects14080711_

Round 1

Reviewer 1 Report

Dear Editors of Insects,

We have conducted a thorough peer review of the manuscript titled "Complex network of Ecdysis triggering hormone signaling in regulating diverse functions during insect development." While the review addresses an important topic in insect biology, we have identified several areas that require improvement. Specifically, the manuscript appears disjointed at times. Perhaps, the different sections were put together or rearranged and lacked a final editorial proofreading. Additionally, the diagrams need to be clarified and provided with detailed legends to accurately represent the data discussed in the main sections. Despite these concerns, we recognize the significance of the research and its potential impact on understanding ETH signaling cascades and their role in diverse insect functions. We recommend that the manuscript undergoes multiple revisions to address the aforementioned issues. Improving the graphic art would be highly recommended. With the necessary improvements, this review will make a valuable contribution to Insects.

Below, we leave several remarks in the order that they appear in the text:

The introduction raises questions that are only addressed later in a somewhat fragmented style, such as the mention of events and hormones. For example, there are gaps between lines 56-57 that are only addressed at lines 113-119. While the authors have the freedom to decide on the style, it may not be the best approach for a review.

It would be beneficial for the reader to include a diagram illustrating the events, tissues, and cell types involved in the phenomena.

Line 64: "Ecdysis triggering hormone" - ETH.

Line 78: "ETHR" - At this point, the abbreviation has not been introduced. It will be introduced at line 82, but without presenting the full form of the abbreviation.

Line 92-95: "C-terminal amidation (a post-translational modification) ecdysis triggering hormone, a linear amidated peptide comprised of 11-34 amino acids, play significant roles in biological functions of ETH." - Please clarify this sentence for better understanding.

Line 96-99: "In M. sexta, the eth gene consists of three exons and two introns. ETH mRNA encodes a 114 amino acid polypeptide (including a 22 amino acid signal peptide) which undergoes processing to produce an 11 amino acid PETH, a 26 amino acid ETH, and another 47 amino acid ETH-associated peptide [8]." Is ETH a product of the same gene as PETH? This should be clearly stated.

Line 113: "CRZ" - Describe the term before introducing the abbreviation.

Line 123: "ETH loses its sensitivity toward CNS" - Please provide clarification on this statement.

Line 125: "based on the insect size" - How does the size of the insect relate to the important variables?

Line 132: "eclosion hormone" - EH.

Line 133-136: "In light of recent research, it is evident that the regulation of ecdysis is under the control of coordinated interaction of several neurohormones, which further adds to the complexity of this process." Later, the authors mention that EH is a neurohormone. At this point, the information was confusing.

Line 138: "The release of ETH from Inka cells is regulated by neurohormones (EH and CRZ) and fluctuating concentrations of the ecdysteroid 20E." - The diagram shows that the release of ETH is regulated by 20E but not EH. The authors need to review the diagram to align with the text description. It is currently not very intuitive, so a descriptive legend should be provided.

Line 140-143: "ETH levels and epitracheal Inka cell sizes increase when 20E reaches peak concentrations, and the release of ETH from Inka cells is triggered by declining concentrations of 20E and the secretion of the brain neuropeptide CRZ." Please clarify when these events occur.

Line 150-151: "At the end of pre-ecdysis, ETH interacts with receptors located on the neurosecretory cells (NSCs) in the brain to initiate the synthesis and release of another brain hormone, EH." It sounds as if EH is being mentioned for the first time, but that is not the case.

Line 152: "EH, the photo-periodically regulated hormone, is necessary for circadian molting behavior in many insects." This is another example where the introduction or presentation of the hormone/molecule is delayed.

Line 153-156: "CZR can release low levels of ETH from Inka cells into the hemolymph and forms a positive feedback loop with EH, which triggers a massive release of both hormones into the hemolymph mediated through elevated levels of Guanosine 3',5'-cyclic monophosphate (cyclic GMP/cGMP)." This information could be visually represented in the diagram.

Line 160: "This EH" - Please revise the sentence for clarity.

Line 161: "CCAP" - Provide the full name for CCAP.

Line 164-165: "When added to the CNS preparation from insects" - What does CNS preparation refer to? Is it homogenates? Which tissues are included?

Line 219-220: "disruption of ETH signaling" - How was this disruption achieved? Was it through gene silencing?

Line 253: "EcR" - The use of the 20E receptor should be consistently referred to as either "EcR" or "20E receptor."

Line 269: "ETH also contributes to JH biosynthesis and activation of the key enzyme JHAMT of JA" - Provide the full name for JHAMT. What does "JA" stand for?

Line 274-275: "βα subunit of the activated G-protein associated with ETHR dissociates." - Please revise the sentence for clarity.

Line 283: "JHAMT, a key enzyme of JH biosynthesis:" - JHAMT has already been introduced as a key enzyme.

Thank you for considering our feedback.

Specifically, the manuscript appears disjointed at times. Perhaps, the different sections were put together or rearranged and lacked a final editorial proofreading. We recommend that the manuscript undergoes multiple revisions to address the aforementioned issues. 

Author Response

We thank the anonymous reviewers for their incisive feedback and all the valuable suggestions and feedback on our manuscript (insects-2499207).  We have addressed all the comments below in more detail. All the weak points and contexts have been corrected and rewritten for improving the quality and clarity of the revised manuscript. We also provided in-depth and additional (required) information in the revised manuscript to improve the continuity and flow the manuscript and making it more informative to the readers. We have also significantly improved the graphic work of all diagrams by providing more in-depth information, clarified and provided with detailed legends to accurately represent the data discussed in the main sections as suggested by the reviewer. We strongly believe that we all these necessary modifications and changes as suggested and recommended by the reviewer the manuscript has been improved immensely and can make a valuable contribution to this esteemed journal.

Reviewer 2 Report

In my opinion, the text, although written in understandable language, does not bring absolutely anything new. It is a summary of knowledge, in the way that have appeared before. In addition, the text is not too extensive and describes only generally given issues. I think that the idea itself was quite good, but it would be worth taking a look at it from a different angle, e.g. by comparing information that concerns insects from the holometabola and hemimetabola groups. Mentioning the second group more than just a sentence that little is known. It would be worth preparing a table for comparisons or figures comparing the differences. Minor remarks? It is worth paying attention to the spelling of insect names, sometimes only the species name appears, and sometimes also the common name. When such a mix takes place in one sentence, it does not look good. It is also worth setting one way of writing for 20E and stick to it, because for the beginners, jumping over the names is quite annoying.

Author Response

We thank the reviewers for their incisive feedback on our manuscript (insects-2499207) all the valuable feedback and suggestions.

Reviewer 2

In my opinion, the text, although written in understandable language, does not bring absolutely anything new. It is a summary of knowledge, in the way that have appeared before. In addition, the text is not too extensive and describes only generally given issues. I think that the idea itself was quite good, but it would be worth taking a look at it from a different angle, e.g., by comparing information that concerns insects from the holometabola and hemimetabola groups. Mentioning the second group more than just a sentence that little is known. It would be worth preparing a table for comparisons or figures comparing the differences.

We thank the reviewers for their incisive feedback on our manuscript (insects-2499207) all the valuable feedback and suggestions.

     We strongly believe that our review article for the first time provides a concrete and novel synthesis of various ETH signaling cascades in diverse insects species (including holo- and hemimetabolous insects), differences in ETH signaling cascades and components among insects and how coordinated actions of various glands and cells/neurons releasing various peptide/neurohormone involved with various networks of ETH signaling cascade that control various phases of ecdysis. We also mentioned in our review that how ETH signaling play crucial role in their development and reproductive regulation of insects during juvenile and adult stages. We also mentioned and expanded the Application of understanding the intricate mechanistic details and various components of ETH signaling cascade in this review article, i.e., how researchers can develop sustainable pest management strategies by identifying novel compounds and targets which appropriate examples in the revised version. We don’t think any other review article published so far covered in such details about molecular/genetic, physiological and applied aspects of ETH signaling network by putting together both past and recent research in this area unlike us. We also strongly believe that our contribution (revised manuscript) will definitely be an excellent addition to the scientific community particular in the area of Insect physiology, Insect Molecular Biology and IPM.

    As per the reviewer’s suggestion, we have also extended the text by providing more in-depth and specific information not only in the text, but also in all the figures and providing extra Tables. Some of the specific areas of ETH signaling Network and applications have been introduced in the revised manuscript. We strongly believe that with comments from all reviewers the revised manuscript has been improved and strengthened immensely. We thank all the reviewers for their comments and feedback.

As per the reviewer’s suggestions, we have introduced a Table (Table S1) in the revised manuscript demonstrating the comparison of regulation of ETH signaling associated with insect development and reproductive success in both holo- and hemimetabolous insects. We thank the reviewer for this excellent suggestion.

Minor remarks

It is worth paying attention to the spelling of insect names, sometimes only the species name appears, and sometimes also the common name. When such a mix takes place in one sentence, it does not look good. It is also worth setting one way of writing for 20E and stick to it, because for the beginners, jumping over the names is quite annoying.

We thank the We have thoroughly edited the manuscript for the spelling errors and corrected in the revised manuscript. We provided both the general names along with their Scientific names and then presented their genus and/or species names throughout the manuscript. 

We are sticking to the 20E throughout the revised manuscript. We thank the reviewer for pointing out these mistakes for improving the manuscript.

Reviewer 3 Report

The review summarizing the network of ETH signalling during insect development is well timed and could be a valuable source for the community, especially if it is written for the broader scientific community. However, in its present state it still needs a tremendous amount of work. The overall structure and flow are hard to follow as it jumps between different concepts. In many cases there is a lack of structure within paragraphs; the paragraphs lack a clear topic sentence and conclusion. The linking together of the different paragraphs to form a full story is also not well developed and in this state the story is very difficult for the wider scientific audience to read and comprehend.

Also the English language needs work as the manuscript  contains several grammatical errors, missing words, and incorrect sentence structures. In addition, sentences are excessively long and complex.

I together with a senior PhD student, have given detailed comments in de file itself. Hopefully the authors can use these to improve the manuscript.

Author Response

We thank the anonymous reviewers for their incisive feedback on our manuscript (insects-2499207), and we have addressed all the comments and concerns below in more detail. We have extensively edited the manuscript based on comments from all the reviewers. We have worked significantly on all the week points, structure of the manuscript, provided all the missing information, significantly improved the graphic work of all diagrams, clarified and provided with detailed legends to accurately represent the data discussed in the main sections. We strongly believe that with all these comments and feedback the revised manuscript has been immensely improved in terms of overall structure and flow of details. We want to sincerely thank the reviewer for providing the detailed feedback.

Round 2

Reviewer 1 Report

Thank you for addressing the concerns I raised in my review. I am glad to see that you have taken the time to make the necessary modifications and that the manuscript is now in an acceptable state for publication.

Although in an acceptable condition, thorough editorial work on the manuscript would be beneficial to correct spelling and minor grammar issues. 

Author Response

Thank you for addressing the concerns I raised in my review. I am glad to see that you have taken the time to make the necessary modifications and that the manuscript is now in an acceptable state for publication.

We sincerely thank the anonymous reviewer for the incisive feedback, all the valuable suggestions and comments on our manuscript. We Strongly believe that all these comments and suggestions from the reviewer helped us improving the manuscript immensely. We are highly grateful to the reviewer for that.  

Comments on the Quality of English Language

Although in an acceptable condition, thorough editorial work on the manuscript would be beneficial to correct spelling and minor grammar issues.

We have thoroughly edited the manuscript for improving the quality of English language of this manuscript. We are also highly grateful to Prof. Dr. Laura Lavine, the Chair for the Department of Entomology at Washington State University and Mr. Chase Baerlocher for their critical feedback, all the edits on our manuscript to improve the quality English Language (grammar, spelling and other errors). We sincerely thank Dr. Laura Lavine and Mr. Chase Baerlocher for critically reading our manuscript and providing 

Reviewer 2 Report

Dear Authors,

Thank you for preparing the corrected text. In this stage it is more complex and will bring more benefits for readers, unless it still needs few corrections.

The table with comparison between hemimetabola and holometabola needs more corrections. The part with developmental stages needs to be delated (this is obvious information). Left information needs to be organised on other way- try to compare, not only write information about. 

Inka cells- Inka needs to written with capital letter. Change on text. 

Author Response

Thank you for preparing the corrected text. In this stage it is more complex and will bring more benefits for readers, unless it still needs few corrections.

We thank the anonymous reviewer for the incisive feedback, all the valuable suggestions and comments on our manuscript. We Strongly believe that all these comments and suggestions from the Reviewer helped us improving the manuscript immensely. We are highly grateful to the reviewer for that.  We have also made some additional minor corrections (highlighted in Red) in revised draft to bring more clarity for the readers. We have also thoroughly edited the manuscript for improving the quality of English language of this manuscript. We are also highly grateful to Prof. Dr. Laura Lavine, the Chair for the Department of Entomology at Washington State University and Mr. Chase Baerlocher for their critical feedback, all the edits on our manuscript to improve the quality of English Language (grammar, spelling and other errors). We sincerely thank Dr. Laura Lavine and Mr. Chase Baerlocher for critically reading our manuscript and providing their feedback.

The table with comparison between hemimetabola and holometabola needs more corrections. The part with developmental stages needs to be delated (this is obvious information). Left information needs to be organised on other way- try to compare, not only write information about. 

We have reorganized and extensively elaborated the Supplementary table S1 to effectively compare of ETH signaling regulation between holo- and hemimetabolous insects in the revised manuscript. In the new Supplementary table in the revised manuscript, we compared the “components and cascades of ETH signaling” between holo- and hemimetabolous insects more effectively by keeping them side by side and comparing each point from both insect groups. We also introduced new information associated with various insect developmental stages in the comparison which we believe will be very helpful for the readers. We are highly grateful to the reviewer for all the excellent excellent suggestions. We strongly believe that our manuscript is highly improved after these two rounds of revisions.

Inka cells- Inka needs to written with capital letter. Change on text. 

We have corrected this issue in the revised manuscript.